# Role of Sulfur Compounds in Garlic as Potential Therapeutic Option for Inflammation and Oxidative Stress in Asthma

**DOI:** 10.3390/ijms232415599

**Published:** 2022-12-09

**Authors:** José L. Sánchez-Gloria, Karla M. Rada, Juan G. Juárez-Rojas, Laura G. Sánchez-Lozada, Ivan Rubio-Gayosso, Fausto Sánchez-Muñoz, Horacio Osorio-Alonso

**Affiliations:** 1Sección de Estudios de Posgrado, Escuela Superior de Medicina, Instituto Politécnico Nacional, Mexico City 11340, Mexico; 2Departamento de Inmunología, Instituto Nacional de Cardiología Ignacio Chávez, Mexico City 14080, Mexico; 3Departamento de Endocrinología, Instituto Nacional de Cardiología Ignacio Chávez, Mexico City 14080, Mexico; 4Departamento de Fisiopatología Cardio-Renal, Instituto Nacional de Cardiología Ignacio Chávez, Mexico City 14080, Mexico

**Keywords:** asthma, inflammation, oxidative stress, garlic, sulfur compounds

## Abstract

Asthma is a chronic inflammatory disease in the airways with a multifactorial origin but with inflammation and oxidative stress as related pathogenic mechanisms. Garlic (*Allium sativum*) is a nutraceutical with different biological properties due to sulfur-containing natural compounds. Studies have shown that several compounds in garlic may have beneficial effects on cardiovascular diseases, including those related to the lungs. Therefore, it is possible to take advantage of the compounds from garlic as nutraceuticals for treating lung diseases. The objective of this article is to review the biological properties of the sulfur compounds present in garlic for the treatment of asthma, as well as the cellular mechanisms involved. Here, we discuss the potential therapeutic effects of garlic compounds in the modulation of inflammation and oxidative stress, as well as its antibiotic and antiviral activities for identifying and testing potential treatment options for asthma management.

## 1. Introduction

The lungs are exposed to the environment and therefore are vulnerable to potentially harmful factors (pathogens, chemicals, etc.) that result in pathological processes, such as neoplasms, infections, and inflammatory/allergic diseases [1]. According to the World Health Organization (WHO), chronic respiratory diseases (CRD) have a heavy economic burden on countries and an adverse impact on national economic development [2]. CRD includes asthma, respiratory allergies, chronic obstructive pulmonary disease (COPD), occupational lung diseases, sleep apnea syndrome, and pulmonary hypertension. It has been reported that more than 300 million people worldwide have asthma (according to the 2016 Global Burden of Disease Study) [3].

Asthma is among the most common CRD, affecting all age groups. Asthma has a multifactorial origin, but its pathophysiology involves the participation and presence of inflammatory mechanisms (an eosinophil-rich airway) and oxidative stress that induce airflow obstruction, mucus hypersecretion, and airway remodeling [4]. In asthma, chronic inflammation leads to an increase of reactive oxygen species (ROS), which translates into remodeling processes that damage the normal architecture of the airways, weakening lung function and, consequently, a worse clinical prognosis [4] (Figure 1). In addition, studies have reported that oxidative stress may be central to the development (cause) and progression (consequence) of airway inflammation. On the other hand, several investigations have shown that supplementation with nutraceuticals with antioxidant properties (e.g., curcumin, zinc, selenium, vitamin D) has a positive effect on the reduction of airway inflammation [5]. Furthermore, nutraceuticals with antioxidant and anti-inflammatory properties could be beneficial in the progression of asthma, as preventive, protective, or therapeutic alternatives.

Currently, asthma is treated with corticosteroids, and patients can also require pharmacological co-treatment. However, corticosteroids cause several adverse effects limiting their application, and patients frequently worsen despite treatments [6]. Therefore, finding more effective therapeutic alternatives with no side effects or toxic properties is necessary.

The evidence has shown that organosulfur compounds present in garlic have beneficial effects on health by modulating inflammation and oxidative stress, which are common pathogenic mechanisms in pulmonary diseases [7,8]. However, studies that summarize the effects of these garlic compounds on asthma are limited. Thus, this review aims to determine the beneficial properties of the sulfur compounds in garlic on asthma and the signaling pathways involved. 

## 2. Association of Oxidative Stress and Inflammation on Asthma

The pathogenesis of asthma involves chronic airway inflammation triggered by allergen agents, ambient temperature, tobacco, bacteria, viruses, respiratory infections, or mechanical stimuli such as exercise. Cells mediating the inflammatory response, such as eosinophils, neutrophils, monocytes, and macrophages, generate ROS and reactive nitrogen species (RNS), contributing to oxidative stress and aggravating the detrimental effects of airway inflammation (Figure 1). On the other hand, oxidative stress in asthma may also be attributed to viral or bacterial infections and is even considered a factor that can exacerbate this pathology [9]. Such a situation was demonstrated in patients with asthma, where a considerable percentage of viral and bacterial co-infections were observed, and the predominant viruses were rhinoviruses, respiratory syncytial virus, influenza virus, and metapneumovirus. At the same time, the major species of bacteria present were *Streptococcus pneumoniae* and *Haemophilus influenzae*. In both cases, the patients had lower exhaled nitric oxide (NO) levels, lower immunoglobulin E (IgE) titers, and a higher incidence of comorbid sinusitis, COPD, or pneumonia [10]. These findings have also been related to an increased risk of hospital readmission. This phenomenon is possible because these allergens regulate the toll-like receptor (TLR) and intracellular adhesion molecule-1 (ICAM-1) pathways, causing an increase in neutrophil degranulation and cell lysis, thus exacerbating the clinical severity of the pathology [11].

Airway inflammation is an essential feature of this pathology where the nuclear factor kappa-light-chain-enhancer of activated B (NF-κB) regulates several genes involved in the immune and inflammatory responses. Here, NF-κB is translocated to the nucleus due to the degradation of inhibitory kappa B (IκB), where NF-κB binds to the promoter region of proinflammatory mediators, including inducible nitric oxide synthase (iNOS), cyclooxygenase 2 (COX2) and matrix metallopeptidase-9 (MMP-9) [12]. Thus, the over-expression of iNOS and COX2 increases NO synthesis, which aggravates the inflammatory response. Chronic inflammation and oxidative stress results in cell proliferation, apoptosis of respiratory epithelium, bronchoconstriction, an increase in mucus secretion and airway remodeling, and, finally, irreversible airflow limitations (Figure 1) [13].

In allergic asthma, exposure to allergens causes an imbalance between the T helper type 1 (Th1) and Th2 cells. The activation of Th2 cells is related to the inflammatory response, leading to tracheal hypersensitivity owing to cytokine release, including interleukin 13 (IL-13), IL-5, and IL-4 [14].

## 3. Garlic

Nowadays, it is indisputable that dietary interventions play a key role in the survival and maintenance of health, as well as in managing diseases, including cardiovascular disease, diabetes mellitus, metabolic syndrome, and cancer [15]. Several studies have reported that nutraceuticals (foods or parts of them) have antioxidant and anti-inflammatory properties that can protect, prevent, or improve chronic diseases such as the lung, including asthma [16]. Garlic is a nutraceutical with health benefits due mainly to the organic sulfur compounds derived from cysteine contained in it [17]. Intact garlic contains non-volatile (non-odiferous and stable) γ-glutamyl-S-alk(en)yl-l-cysteines, namely, γ-glutamyl-S-allyl-l-cysteine, γ-glutamyl-S-trans-1-propenyl-l-cysteine, and S-alk(en)yl-l-cysteine sulfoxides such as S-allyl-l-cysteine sulfoxide (alliin), S-(trans-1-propenyl)-l-cysteine sulfoxide (isoalliin), and S-methyl-l-cysteine sulfoxide (methiin) with a small amount of S-allyl cysteine (SAC). When garlic cloves are crushed or chopped, the enzyme alliinase stored in the vacuoles is released, which encounters cytosolic alliin to convert it into a series of thiosulfinates, most notably allicin (diallyl thiosulfinate). The highly reactive, unstable, and volatile allicin breaks down to produce a large number of sulfides, which are oil-soluble compounds responsible for garlic’s characteristic odor and taste. These sulfides correspond to diallyl sulfide (DAS), diallyl disulfide (DADS), diallyl trisulfide (DATS), methyl allyl disulfide (MADS), methyl allyl sulfide (MAS), ajoene, and vinyl dithiins (2-vinyl-1,3-dithiin, 3-vinyl-1,2-dithiin). The water-soluble garlic compounds are SAC, S-allylmercapto-l-cysteine (SAMC), and S-methyl cysteine [8].

## 4. Effect of Garlic Compounds on Asthma

A study investigated the effects of intraperitoneal injection of aged garlic extract (AGE) on established allergic airway inflammation in a murine model (BALB/c mice) [18]. The injection of AGE caused a decrease in the allergic airway inflammation, including eosinophil percentage in bronchoalveolar lavage fluid (BALF), immunoglobulin G_1_ (IgG_1_) levels in BALF and serum, the proportion of mucous-producing goblet cells, and peribronchial and perivascular inflammation. It also increased BALF’s interferon-gamma (IFN-γ) levels. The results suggested that AGE could attenuate inflammatory features of allergic airway inflammation. 

Shin et al. investigated the effects of DADS on airway inflammation using a mouse model of ovalbumin-induced asthma [19]. In this study, DADS suppressed the expression of iNOS, COX2, and MMP-9, which decreased NF-κB activation, thus inhibiting the production of inflammatory markers (IL-1β and IL-6) in experiments in vitro and in vivo. The suppressed expression of MMP-9 by DADS treatment caused a reduction in IL-4, IL-5, IL-13, and IgE in the lung tissue of rats with asthma. In addition, ovalbumin-induced asthma decreased the expression of IFN-γ levels, while DADS treatment significantly increased the expression of IFN-γ and the expression of antioxidant proteins, such as nuclear factor (erythroid-derived 2)-like 2 (Nrf2) and hemeoxygenase-1 (HO-1) in experiments in vivo and in vitro, leading to reduced ROS production [19]. Thus, the results showed that DADS decreases the inflammatory response by enhancing the antioxidant status induced by Nrf2 activation.

Other studies investigated the effects of DAS orally administered on ovalbumin-induced pulmonary inflammation of asthma mice [20]. In this study, DAS decreased airway inflammation, mucus secretion, and oxidative damage in the lung of asthma mice. In asthmatic responses, Nrf2 disruption causes an increase in the levels of Th2 cytokines IL-4 and IL-13. DAS administration elevated the Nrf2 translocation from the cytosol to the nucleus in the lung cells and consequently decreased the inflammatory state. Specifically, DAS reduced the number of eosinophils in BALF, preventing inflammatory cell infiltration and the generation of Th2 cytokines (IL-4 and IL-10) via Nrf2. Furthermore, DAS decreased the expression of 8-Hydroxy-2’-deoxyguanosine (8-OHdG) and 8-isoprostane, two biomarkers of oxidative damage, suggesting that DAS reduced ROS generation and prevented oxidant-induced damage in the lung [20]. In addition, DAS treatment downregulated the expression of microRNAs (miRNAs) such as miR-144, miR-34a, and miR-34b/c, which play a role in oxidant and inflammatory activities [21,22]. 

Inhalant allergens such as dust or house mites are considered the most important source of allergens worldwide. Dust mites, particularly *Dermatophagoides pteronyssinus* (Der p), constitute one of the most critical risk factors for allergic respiratory diseases in patients with a genetic predisposition. In this context, the oral administration of the water-soluble fraction of garlic (collected in Taichung City, Taiwan) on Der p-induced allergic airway inflammation in mice was evaluated [23]. The total inflammatory cells determined in the lung of asthmatic mice were increased by Der p; however, garlic treatment inhibited the total cell counts and inflammatory cell infiltration (eosinophils and lymphocytes) around perivascular space. These results are supported by the reduction of Th2 cytokines by garlic treatment, an important cytokine that regulates the secretion of IgE. In contrast, the garlic extract fraction was found to increase IFN-γ levels. These results indicate that garlic reduces airway inflammation by decreasing Th2 cytokines and increasing Th1 cytokines. In addition, IgE is related to the NF-κB activation [24]. The reduction of IgE by the garlic fraction treatment also inhibited the phosphorylation of NF-κB and the decrement in the expression of IL-13 and IL-4. Garlic extract administration modulated the anti-inflammatory response by inhibiting the IL-6/PI3K/Akt/NF-κB pathway. 

Hsieh et al. also reported that garlic extract (collected in Taichung City, Taiwan) corrected the imbalance of Th1 and Th2 cells in BALB/c mice with Der p-induced asthma. Again, Th2 cytokines (IL-4, IL-5, and IL-13) decreased with garlic administration, causing a reduction in the stimulation of mucus secretion from epithelial cells in the airways, decreasing the expression of vascular endothelial cell adhesion molecules (VCAM) and inhibiting IgE production. On the other hand, it increased IFN-γ and IL-12 and restored the expression of IL-10 in BALF [23]. SAC is another compound in garlic that has been reported to have beneficial effects on the ovalbumin-induced asthma model [17]. In this study, SAC attenuated airway hyperresponsiveness and inflammatory cell infiltration by significantly reducing inflammatory cell counts. In asthma, goblet cells show an increase in mucin 5AC (MUC5AC), a major component of airway mucus, caused by an increase in Th2 cytokines. Mucus production from goblet cells is triggered by an increase in Th2 cytokines and increased activity of molecules in the inflammatory signaling pathway, such as NF-κB [25]. In addition, the oral administration of SAC decreased Th2 cytokines and IgE levels in BALF and serum. Finally, SAC administration inhibited NF-κB translocation to the nucleus and, thus, the transcription of inflammatory proteins, resulting in reduced airway hyperresponsiveness and MUC5AC. 

Other studies have reported that oral administration of SAC has beneficial effects in neonatal asthmatic rats (in an ovalbumin-induced asthmatic animal model) [26]. SAC administration decreased the inflammation and the infiltration of eosinophils, lymphocytes, mast cells, and monocytes, as well as the number of goblet cells in the airway. In addition, SAC decreased smooth muscle mass, mucous gland hypertrophy, and vascular congestion in the asthmatic model. In this study, the administration of SAC decreased the expression of fibrinogen, prothrombin, and thrombin time; these parameters are related to the degree of inflammation in asthma. Moreover, SAC administration decreased the expression of TNF-α, IL-1β, IL-6, IL-13, and IL-17. In contrast, the expression levels of IL-10 were increased with the treatment [26]. Studies have reported that the production of several proinflammatory cytokines is inhibited in the presence of prostaglandin E2 (PGE2). PGE2 suppresses the production of the Th1 cytokine secretion [27]. However, PGE2 can act on uncommitted B lymphocytes to promote isotype switching to IgE or IgG1 [28]. IL-6 is a proinflammatory mediator involved in synthesizing PGE2 and the infiltration of eosinophils in the airway. In asthmatic animals, high expression of COX2 correlates with IL-6, which regulates immune cells to generate PGE2. SAC treatment decreased IL-6, PGE2, and COX2. In addition, other compounds derived from arachidonic acid, such as leukotriene, cysteinyl leukotrienes, and leukotrienes B4 in eosinophils, act as potent bronchoconstrictor and cause airway smooth muscle constriction and increase mucus secretion. These effects were decreased by SAC treatment [26]. IL-13 also plays a role in this pathology by regulating the inflammation and remodeling of the lung tissues by Th2 cytokines. This mechanism is accompanied by a coordinated response of the various chemokines, such as eotaxin, which, upon activation, regulated normal T cell expressed and secreted macrophage inflammatory protein-1 beta (MIP1-β) and monocyte chemoattractant protein-1 (MCP-1), resulting in the Th2 inflammatory response in the lungs. This phenomenon increases the traffic of eosinophils from the bloodstream to the airways, increasing adhesion molecules to join the epithelial cells of the airways. In this study, IL-13 inhibition through SAC administration could ameliorate asthma-related inflammatory events by downregulating Th2 cytokines. Therefore, SAC may be an important nutraceutical for inhibiting airway inflammation by decreasing the expression of inflammatory cytokines in asthma patients. The garlic compounds could also be used as a coadjutant or therapeutic option in treating pathogen-infected asthma exacerbation patients, mainly bacterial and viral infections. Based on this knowledge, the dietary intake of these nutraceuticals as coadjuvant therapy could reduce the adverse effects of antiviral drugs, including preventing bacterial or viral infections that exacerbate asthma. In this context, the antimicrobial effects of allicin on clinical isolates of pathogenic lung bacteria from the genera *Pseudomonas*, *Streptococcus*, and *Staphylococcus*, including multi-drug resistant (MDR) strains, were demonstrated. Thus, allicin inhibited the growth of most *Pseudomonas*, *Streptococcus*, and *Staphylococcus* isolates; moreover, allicin provided more therapeutic advantages than conventional antibiotics (erythromycin and clindamycin) [29]. Furthermore, allicin was shown to have a synergistic effect with oral antibiotics, suggesting a valuable addition to available treatments for lung infections such as asthma [30]. Concerning the antiviral effects of garlic compounds, a recent study assessed the effect of DATS on H9N2 avian influenza virus infection; these viruses cause pulmonary edema, infiltration of inflammatory cells and cytokine secretion (IL-6 and TNF-α), interstitial and alveolar edema, and hemorrhage. DATS reduced the pathological changes of H9N2-induced infection and the expression of IL-6 and TNF-α, in contrast, increased the expression of antiviral cytokines [retinoic acid-inducible gene I (RIG-1) and IFN-β]. These effects were related to decreased lung viral load [31]. Therefore, it is suggested that DATS may be an agent that enhances innate immunity against H9N2 avian influenza virus infection due to its antiviral activity [31]. Clinical studies in patients with the common cold who were given daily supplements of garlic capsules containing allicin showed a reduction in viral infection and reinfection. Such an effect may be due to the immunomodulatory effects of garlic, specifically allicin [32]. Other studies have reported that the organosulfur compounds in garlic showed antiviral activity against viruses that cause respiratory infections, such as influenza, parainfluenza, coronavirus, rhinovirus, and adenovirus [33]. Over the years, these viruses have been found to be associated with about 80–85% of asthma exacerbations in school-age children (9–11 years old) [34]. A recent study showed that allicin exerts antiviral effects when applied to cell cultures damaged by SARS-CoV-2, thus decreasing more than 60% of the infectious viral particles and viral RNA of SARS-CoV-2. In addition, allicin prevented changes in the proteome caused by SARS-CoV-2 infection [35].

## 5. Discussion

This work reviews the biological properties of the sulfur compounds present in garlic for the treatment of asthma. The effects of the administration of AGE and garlic extract, the oil-soluble compounds such as DAS, DADS, DATS, and allicin, and water-soluble compounds such as SAC were reviewed. The garlic compounds showed several biological functions, including anti-inflammatory, antioxidant, and antibiotic activities. To the best of our knowledge, this is the first review to explore the molecular mechanisms through which the sulfur compounds in garlic have beneficial effects on lung disease, mainly asthma.

Asthma is a chronic inflammation characterized by several factors that trigger an immune response; these include infections and genetic and environmental factors. The modulation of inflammation is key to controlling and preventing the disease’s aggravation and the inhalation or systemic medication using corticosteroids is the first-line medicine of defense for the treatment of asthma. The economic cost of asthma is high in severe or uncontrolled asthma, involving drug costs and hospital admissions. In low- and middle-income countries, asthma significantly affects daily activities, life, and family finances [2]. In addition, the administration of these drugs may trigger the risk of adverse events. For example, corticosteroid administration by inhalation induces the risk of pulmonary and oropharynx fungal infection, while the topic application can induce systemic immunity impairment, infection, steroid diabetes, and osteoporosis [36].

The publication “Global surveillance, prevention and control of chronic respiratory diseases: a comprehensive approach” mentions that despite underestimating the potential use of complementary and alternative medicine, its use is fairly routine in low and middle-income countries, being extremely important for the care of the sick [37]. This therapeutic alternative was promoted for the first time at the 55th World Health Assembly [2]. Thus, traditional medicine is often the first step in disease management, mainly due to patient beliefs and the high cost of drugs; such factors have positively impacted the promotion of traditional medicine. However, traditional medicine must be accompanied by validated modern medicine. Likewise, in this review, we showed that the sulfur compounds in garlic were non-toxic (low toxicity) or did not have side effects in the asthma models. In this sense, consuming these active biomolecules in one’s diet should be considered as a potential therapeutic option or a coadjutant to care for these pathologies [38]. This study showed that garlic’s sulfur compounds decreased mucus secretion, pulmonary wall thickening, and cell infiltration. In general, sulfur compounds decreased inflammatory events by decreasing Th2 cytokines (IL-4, IL-5, and IL-13) and immunoglobulin IgE and IgG1 (Figure 2). In addition, these compounds caused the inhibition of NF-κB and the subsequent genes involved in the inflammatory responses (iNOS, COX2, MMP-9, IL-1β, IL-6, and TNF-α) and various chemokines mentioned above. In contrast, sulfur compounds increased IFN-γ, IFN-β, IL-10, and IL-12 levels, enhancing innate immunity and thus playing a pivotal role in the expression of Th1 cytokines and treating allergic respiratory diseases in patients with a genetic predisposition.

In inflammation and oxidative stress, sulfur compounds play a crucial role. The endogenous sulfur compounds are mainly represented by cysteine, methionine, hydrogen sulfide (H_2_S), and glutathione, these last two molecules with crucial biological activity as vasodilators and antioxidants, respectively [39,40]. On the other hand, the sulfur garlic compounds such as DAS, DADS, DATS, MADS, MAS, ajoenes, vinyl dithiins, SAC, SAMC, and S-methyl cysteine are degraded or metabolized and can act as donors of both molecules H_2_S or glutathione [41,42]. Therefore, sulfur garlic compounds act as an exogenous supplier of the precursors of reduced glutathione (GSH) and H_2_S. As we showed in this work, sulfur-containing compounds protect the lungs from chronic diseases such as chronic obstructive pulmonary disease and asthma. In respiratory diseases and asthma, cytokines promote the survival of inflammatory cells [43,44], contributing to the maintenance of chronic inflammation and producing proinflammatory cytokines such as IL-1β, IL-6, and TNF-α, which are increased in the sputum and BALF of patients with asthma [44]. Additionally, the inflammatory response in asthma increases oxidative stress aggravating the disease. Thus, garlic compounds can act as direct antioxidants via the formation of GSH and through scavenging free radicals and by an indirect mechanism via stimulation of the master antioxidant system Nrf2/Keap1, contributing to an increase in the expression of endogenous antioxidant enzymes, as well as the downregulation of miRNAs related with oxidative stress (Figure 3).

On the other side, there is a close relationship between oxidative stress and inflammation; thus, between Nrf2 and NF-κB. Nrf2 regulates the inflammation NF-κB mediated at various levels. Because of the oxidative stress decrement, the activation of NF-κB induced by oxidant stress is blocked [45]. Moreover, it has been suggested that Nrf2 may also modulate inflammation. Through the prevention of the proteasomal degradation of IκB, Nrf2 can maintain the NF-κB/IκB complex; therefore, this effect contributes to inhibiting the translocation of NF-κB to the nucleus, blocking the transcription of proinflammatory cytokines [46] and infiltration of inflammatory cells in airways. Another anti-inflammatory mechanism is associated with the inhibition of IkB degradation mediated by the increased expression of Nrf2-induced antioxidant enzymes [47]. Furthermore, as we described in this manuscript, garlic compounds have been shown to have anti-inflammatory effects via modulation of the expression of NF-κB/IκB (Figure 3).

On the other hand, garlic compounds showed therapeutic potential for treating asthma induced by bacterial, viral, and inhalant allergens agents such as house mites. Such outcomes resulted from its effects through inhibition of bacterial growth, reduction of viral load, and modulation of the inflammatory response, thus causing a decrease in the stimulation of mucus secretion from epithelial cells in the airways and attenuating lung damage (Figure 2). A molecular docking analysis showed that garlic-derived sulfur compounds might have antiviral potential to prevent COVID-19, a disease that induces multi-organic damage but affects pulmonary structure and function [48]. For this reason, infections by viruses and bacteria are an essential issue in the care of these asthmatic patients because they are considered factors that exacerbate the damage. Thus, the treatment should also be directed toward these organisms.

In addition, it is well known that the sulfur compounds present in garlic are highly reactive to thiol groups present in proteins [49]. During exposure to allicin or other organosulfur compounds, proteins containing catalytically important thiol groups are oxidized and inhibited [50]. Studies have reported that compounds in garlic inhibit viral integrin signaling pathways to block virus entry into host cells [51]. Integrins are transmembrane receptor proteins that respond to the adhesion or binding of the extracellular matrix of the cell [52]. Therefore, the antiviral effects of garlic and its sulfur compounds arise through various mechanisms, such as blocking viral entry, fusion in host cells, inhibition of viral replication, and the enhancement of the host’s immune response.

Similarly, the antibacterial activity of garlic and its compounds arises through a reaction with the free sulfhydryl group on proteins and/or enzymes to inactivate them. Garlic and its compounds alter the composition and integrity of the bacterial cell membrane and/or cell wall. Additionally, it has been proposed that compounds in garlic could influence DNA, RNA, and protein synthesis (Figure 3) [53]. Interestingly, these antibacterial mechanisms of garlic and its compounds have been observed in gram-positive and gram-negative bacteria, suggesting that these nutraceuticals have a similar beneficial effect on both groups of bacteria [50]. However, the disadvantage of the activity of these compounds is that it is not specific, which could restrict their clinical application, in addition to being unstable compounds [54]. Thus, future research should provide better stability to this compound since it is known that allicin is unstable and rapidly decomposes into other sulfur compounds (polysulfanes, ajoene, etc.) during heating [55]. Furthermore, most of the allicin is degraded to 2-propenethiol and allyl methyl sulfide and excreted, whereas, in the blood, the effective dose of allicin is reduced by its reaction with glutathione [49].

The pieces of evidence suggest that sulfur garlic compounds are helpful in the management of asthma. The protective mechanisms include vasodilation, antibacterial, antiviral, antioxidant, and anti-fibrotic activities, which contribute to the modulation of the inflammatory response in asthma (Figure 3). Thus, the sulfur compounds in garlic may be helpful as a therapeutic option or coadjutant for treating asthma of different causes, including genetic predisposition. However, more research is necessary in order to (a) identify the doses that have beneficial effects in patients with lung diseases, (b) the side effects of the consumption of sulfur compounds present in garlic, and (c) the possibility of using these nutraceuticals as coadjutants in the cure of this pathology. A synergistic action between garlic compounds and corticosteroids, antibiotics, bronchodilators, or anti-inflammatories must be considered; in the case of an allergy to a particular antibiotic or drug, garlic compounds might also be pondered as an alternative therapy.

We hope this review will help to inform the mechanisms through which sulfur compounds exert their beneficial effects on lung diseases focusing on asthma. A weakness of this review is that we did not focus on evaluating the side effects, toxicity, safety, and effectiveness of these treatments with sulfur compounds so they can be used in clinical research. However, it has been reported that garlic compounds could be a therapy with fewer side effects [8].

## 6. Conclusions

Asthma is a multifactorial lung disease where oxidative stress and inflammation play a key role in development and progression. Therefore, the modulation of these factors is essential to control the disease. The sulfur compounds in garlic, through antioxidant, antibacterial, and antiviral mechanisms and the modulation of inflammation and antiviral cytokines, offer simultaneous and diverse benefits that delay the progression of the disease and thus can be considered as coadjutants or therapeutic alternatives in asthma treatment.

## Figures and Tables

**Figure 1 ijms-23-15599-f001:**
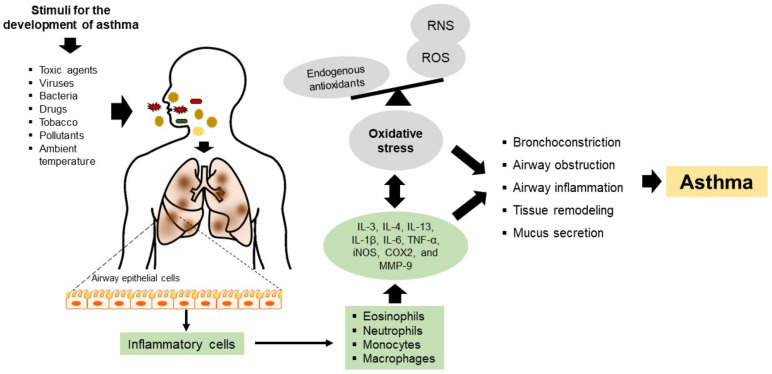
Pathophysiology mechanisms of asthma. Toxic agents, viruses, bacteria, drugs, tobacco, pollutants, and ambient temperature damage airway epithelial cells, inducing the recruitment of inflammatory cells such as eosinophils, neutrophils, monocytes, and macrophages. The inflammatory cells produce cytokines, chemokines, and substances that contribute to local and systemic inflammatory events that aggravate and cause inflammation and oxidative stress in the airways. Thus, chronic inflammation in asthma is accompanied by increased ROS production; however, this imbalance may also contribute to the induction of airway inflammation. Oxidative stress can cause or be a consequence of asthma and even exacerbate this condition. Th2: T helper cells type 2; IL: interleukin 1β, 3, 4, 6, and 13; TNF-α: tumor necrosis factor-alpha; ROS: reactive oxygen species; RNS: reactive nitrogen species; MMP-9: matrix metalloproteinase-9; iNOS: inducible nitric oxide synthase; and COX2: cyclooxygenase 2.

**Figure 2 ijms-23-15599-f002:**
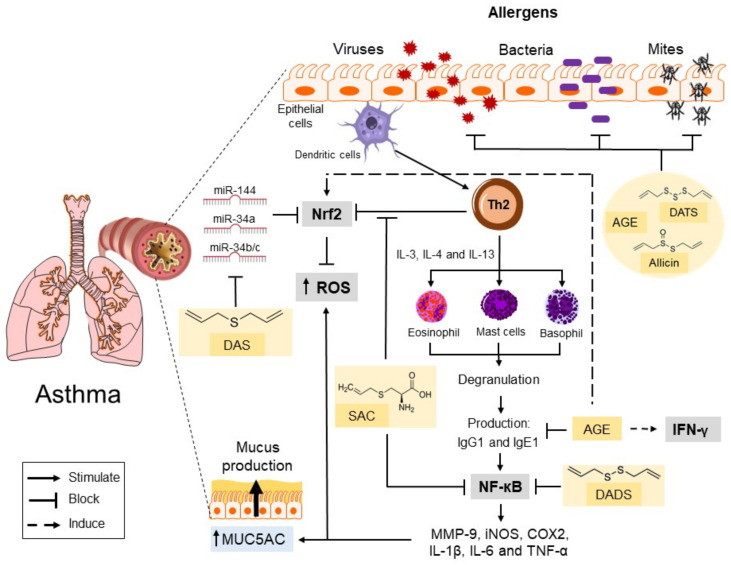
Effects of AGE, DADS, DAS, SAC, DATS, and allicin on asthma. Aged garlic extract and sulfur compounds decreased Th2 cytokines, NF-κB, and the subsequent infiltration of inflammatory cells in the lung. The mechanisms responsible for garlic compounds’ beneficial effects on asthma include antibiotic, antiviral, anti-inflammatory, and antioxidant activities attenuating lung damage. The nutraceuticals in garlic reduce ROS production by raising the expression of antioxidant proteins such as Nrf2. In addition, sulfur compounds downregulated the expression levels of miRNAs related with oxidative stress and inflammation. AGE: aged garlic extract; DADS: diallyl disulfide; DAS: diallyl sulfide; SAC: S-allyl cysteine; DATS: diallyl trisulfide; Th2: T helper cells type 2; IL: interleukin 3; IgG1: Immunoglobulin G1; IgE1: Immunoglobulin E1; NF-κB: nuclear factor kappa B; MMP-9: matrix metalloprotease-9; iNOS: inducible nitric oxide synthase; COX2: cyclooxygenase 2; ROS: reactive oxygen species; MUC5AC: Mucin 5AC; and miR: microRNAs.

**Figure 3 ijms-23-15599-f003:**
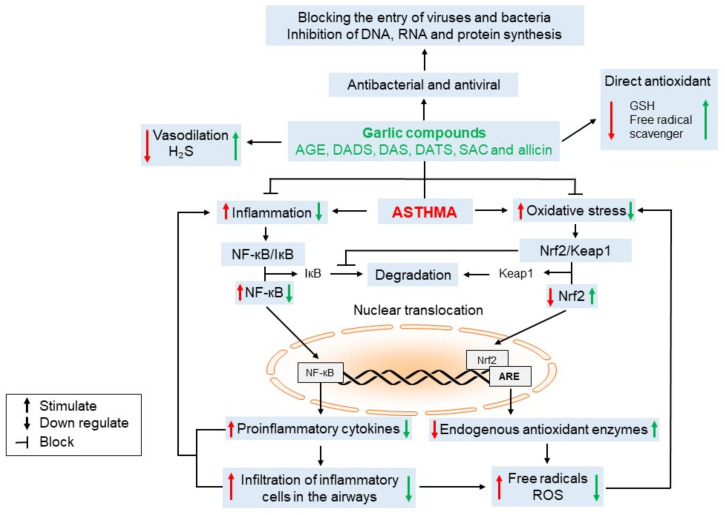
Cellular mechanisms from sulfur garlic compounds involved in the modulation of pathogenic pathways in asthma. Abbreviations: ARE: antioxidant response element; DAS: diallyl sulfide; DADS: diallyl disulfide; GSH: reduced glutathione; H_2_S: hydrogen sulfide; IκB: inhibitor of kB; Keap1: Kelch-like ECH-associated protein; NF-κB: nuclear factor kappa-light-chain-enhancer of activated B; Nrf2: NF-E2 p45-related factor 2; ROS: reactive oxygen species; SAC: S-allyl cysteine. Red arrows indicate the effects induced by asthma, while green arrows indicate the effects induced by the sulfur compounds in garlic.

## Data Availability

Not applicable.

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
