# Peer review of "Role of Sulfur Compounds in Garlic as Potential Therapeutic Option for Inflammation and Oxidative Stress in Asthma"

_ijms, 2022, doi:10.3390/ijms232415599_

Round 1
Reviewer 1 Report
It is a very clear review on the effect of garlic on asthma. Logically built, well documented material. The only part I miss and suggest to incorporate is a detailed paragraph on endogenous and exogenous sulfur products with their biology, production and influence on allergic pathways.
Author Response
México City, November 11th, 2022
Ms. Tinsley Qiu
Assistant Editor
International Journal of Molecular Sciences
Enclosed, please find the revised version of the manuscript ijms-2005963 “Role of sulfur compounds in garlic as potential therapeutic option for inflammation and oxidative stress in asthma” by José Luis Sánchez-Gloria, Karla M. Rada, Juan Gabriel Juárez-Rojas, Laura Gabriela Sánchez-Lozada, Ivan Rubio-Gayosso, Fausto Sánchez-Muñoz, Horacio Osorio-Alonso.
This major manuscript revision includes a point-by-point response to the comments raised during the review.
As the Reviewer suggested, we have discussed the influence of sulfur compounds in garlic on asthma and the antimicrobial and antiviral mechanisms. Furthermore, we have included a figure with the leading cellular mechanisms of the sulfur garlic compounds on asthma.
We greatly appreciate the careful review of our work. We believe that the observations and suggestions significantly improved the quality of the manuscript.
Thanks for the opportunity to re-submit our paper.
Regards,
Horacio Osorio Alonso, Ph.D.
For all authors
Comments and Suggestions for Authors
It is a very clear review on the effect of garlic on asthma. Logically built, well documented material. The only part I miss and suggest to incorporate is a detailed paragraph on endogenous and exogenous sulfur products with their biology, production and influence on allergic pathways.
Answer: We thank the reviewer comment´s:
According to the Reviewer’s suggestion, we have added a paragraph in the discussion section and an additional figure.
“In asthma context, sulfur compounds play a crucial role. The endogenous sulfur compounds are represented mainly by cysteine, methionine, H2S, and glutathione, these last two molecules with important biological activity as vasodilators and antioxidants respectively [40,41]. On the other hand, the sulfur garlic compounds such as DAS, DADS, DATS, MADS, MAS, ajoenes, vinyl dithiins, SAC, SAMC, and S-methyl cysteine through its metabolism can be donors of both molecules [42-44], therefore acting as exogenous suppliers of the precursors of GSH and H2S.
As we showed in this work, the sulfur-containing compounds protect the lungs from chronic diseases such as chronic obstructive pulmonary disease and asthma. In respiratory diseases and asthma, cytokines promote the survival of inflammatory cells [45,46], contributing to the maintenance of chronic inflammation and producing proinflammatory cytokines such as IL-1β, IL-6, and TNF-α, which are found in increased amounts in the sputum and BALF of patients with asthma [46]. Additionally, the inflammatory response in asthma increases oxidative stress aggravating the disease. In this sense, the garlic compounds can act as antioxidants direct via the formation of GSH and through scavenging free radicals and by an indirect mechanism via stimulation of the master antioxidant system Nrf2/Keap1 contributing to an increase in the expression of endogenous antioxidant enzymes (Figure 3).
On the other side, there is a close relationship between oxidative stress and inflammation, thus between NRf2 and NF-кB. Nrf2 regulates the inflammation NF-кB-mediated at various levels. Because of the decrement of oxidative stress, the activation of NF-кB induced by oxidant stress is blocked [47]. Besides, it has been described that the inflammation also may be modulated by Nrf2. Through the prevention of the proteasomal degradation of IkB, Nrf2 can maintain the complex NF-кB/IкB, therefore, this effect contributes to inhibiting the translocation of NF-кB into the nucleus blocking the transcription of proinflammatory cytokines [48]. Another anti-inflammatory mechanism is associated with the inhibition of IкB degradation mediated by the increased expression of antioxidant enzymes Nrf2-induced [49] Furthermore, as we described along the manuscript the garlic compound has been shown anti-inflammatory effects via modulation of expression of NF-кB/IкB (Figure 3).
The protective mechanisms include vasodilation, antibacterial, antiviral, antioxidant, and anti-fibrotic activities, which together contribute to modulating the inflammatory response in asthma. Thus, the sulfur compounds in garlic may be useful as a therapeutic option or coadjutant for the treatment of asthma of different causes including the genetic predisposition.”
Additionally, we have submitted to review the grammar.

Reviewer 2 Report
Dear Authors,
This review "Role of sulfur compounds in garlic as potential therapeutic option for inflammation and oxidative stress in asthma" has been a thoroughly fascinating read and the authors should be commended on it. Overall well structured and adequately addresses the importance of the sulfur compounds in garlic. However, as alluded to within the review, asthma is a disease of multifactorial origins and in the present form, the review very comprehensively covered some aspects of asthma but only very briefly touched on the antimicrobial and antiviral effects of the sulfur compounds. Despite listing the limitation of the review, it is also important that they be discussed, in order to provide a comprehensive, non-biased review on the properties of the sulfur compounds as a therapeutic for asthma.
Author Response
México City, November 11th, 2022
Ms. Tinsley Qiu Assistant Editor
International Journal of Molecular Sciences
Enclosed, please find the revised version of the manuscript ijms-2005963 “Role of sulfur compounds in garlic as potential therapeutic option for inflammation and oxidative stress in asthma” by José Luis Sánchez-Gloria, Karla M. Rada, Juan Gabriel Juárez-Rojas, Laura Gabriela Sánchez-Lozada, Ivan Rubio-Gayosso, Fausto Sánchez-Muñoz, Horacio Osorio-Alonso.
This major manuscript revision includes a point-by-point response to the comments raised during the review.
As the Reviewer suggested, we have discussed the influence of sulfur compounds in garlic on asthma and the antimicrobial and antiviral mechanisms. Furthermore, we have included a figure with the leading cellular mechanisms of the sulfur garlic compounds on asthma.
We greatly appreciate the careful review of our work. We believe that the observations and suggestions significantly improved the quality of the manuscript.
Thanks for the opportunity to re-submit our paper.
Regards,
Horacio Osorio Alonso, Ph.D.
For all authors
Comments and Suggestions for Authors
Dear Authors,
This review "Role of sulfur compounds in garlic as potential therapeutic option for inflammation and oxidative stress in asthma" has been a thoroughly fascinating read and the authors should be commended on it. Overall well structured and adequately addresses the importance of the sulfur compounds in garlic. However, as alluded to within the review, asthma is a disease of multifactorial origins and in the present form, the review very comprehensively covered some aspects of asthma but only very briefly touched on the antimicrobial and antiviral effects of the sulfur compounds. Despite listing the limitation of the review, it is also important that they be discussed, in order to provide a comprehensive, non-biased review on the properties of the sulfur compounds as a therapeutic for asthma.
Answer: We thank the reviewer comment´s:
According to the Reviewer's suggestion, we added a paragraph in section: 4. Effect of garlic compounds on asthma.
“Based on this knowledge, the dietary intake of these nutraceuticals as coadjuvant therapy could reduce the adverse effects of antiviral drugs including the prevention of bacterial or viral infections that exacerbate asthma.
In this sense, a group of researchers evaluated the antibiotic effect of allicin vapor on lung pathogenic bacteria cell lines. In this study, allicin vapor inhibited Pseudomonas, Streptococcus, and Staphylococcus bacterial growth. Furthermore, allicin was shown to have a synergistic effect with oral antibiotics, suggesting a valuable addition to available treatments for lung infections such as asthma [31].
Clinical studies in patients with the common cold who were given daily supplements of garlic capsules containing allicin showed a reduction in viral infection and reinfection. This may be due to the immunomodulatory effects of garlic, specifically allicin [33]. However, asthma is a more severe disease with more clinical conditions of attention. Other studies have reported that the organosulfur compounds of garlic showed antiviral activity against viruses that cause respiratory infections, such as influenza, parainfluenza, coronavirus, rhinovirus, and adenovirus [34]. Over the years, these viruses have been found to be associated with about 80-85% of asthma exacerbations in school-age children (9-11 years old) [35]. A recent study showed that allicin exerts an antiviral effect when applied to cell cultures damaged by SARS-CoV-2. Allicin decreased more than 60% of the infectious viral particles and viral RNA of SARS-CoV-2. Also, changes in the proteome caused by SARS-CoV-2 infection were observed by allicin [36]. “
Additionally, we included in the discussion section the following:
“In addition, it is well known that the sulfur compounds present in garlic are highly reactive to thiol groups present in proteins [51]. During exposure to allicin or other organosulfur compounds, proteins containing catalytically important thiol groups are oxidized and inhibited [52]. Studies have reported that compounds in garlic inhibit viral integrin signaling pathways to block virus entry into host cells [53]. Integrins are transmembrane receptor proteins that respond to the adhesion or binding of the extracellular matrix of the cell [54]. Therefore, the antiviral effects of garlic and its sulfur compounds arise through various mechanisms, such as blocking viral entry, fusion in host cells, inhibition of viral replication, and the enhancement of the host's immune response.
Similarly, the antibacterial activity of garlic and its compounds arises through a reaction with the free sulfhydryl group on proteins and/or enzymes to inactivate them. Garlic and its compounds alter the composition and integrity of the bacterial cell membrane and/or cell wall. Additionally, it has been proposed that compounds in garlic could influence DNA, RNA, and protein synthesis [55]. Interestingly, these antibacterial mechanisms of garlic and its compounds have been observed in both gram-positive and gram-negative bacteria, suggesting that these nutraceuticals have a similar beneficial effect on both groups of bacteria [56]. However, the disadvantage of the activity of these compounds is that it is not specific, which could restrict their clinical application, in addition to being unstable [57].
Thus, future research should provide improved stability to this compound since it is known that allicin is unstable and rapidly decomposes into other sulfur compounds (polysulfanes, ajoene, etc.) during heating [58]. Furthermore, in the acidic pH of the stomach, most of the allicin is degraded and excreted to 2-propenethiol and allyl methyl sulfide, whereas, in the blood, the effective dose of allicin is reduced by its reaction with glutathione [51].

Round 2
Reviewer 2 Report
Dear Authors,
Thank you once again for providing such as interesting read. The revised version reads much better and more comprehensive after the inclusions of the previous recommendations. Just a very minor comment on ensuring that sentence structure, grammar and tenses are appropriate as there were a few sentences throughout the manuscript that came across slightly disjointed and did not flow as well.
Author Response
México City, November 17th, 2022.
Ms. Tinsley Qiu
Assistant Editor
International Journal of Molecular Sciences
Enclosed, please find the revised version of the manuscript ijms-2005963 “Role of sulfur compounds in garlic as potential therapeutic option for inflammation and oxidative stress in asthma” by José Luis Sánchez-Gloria, Karla M. Rada, Juan Gabriel Juárez-Rojas, Laura Gabriela Sánchez-Lozada, Ivan Rubio-Gayosso, Fausto Sánchez-Muñoz, Horacio Osorio-Alonso.
This minor manuscript revision includes a point-by-point response to the comments raised during the review.
We have carefully reviewed the manuscript and have restructured some sentences as appropriate so that there is connection and coherence in the content of the article.
We greatly appreciate the careful review of our work. We believe that the observations and suggestions significantly improved the quality of the manuscript.
Thanks for the opportunity to re-submit our paper.
Regards,
Horacio Osorio Alonso, Ph.D.
For all authors
Comments and Suggestions for Authors
Dear Authors,
Thank you once again for providing such as interesting read. The revised version reads much better and more comprehensive after the inclusions of the previous recommendations. Just a very minor comment on ensuring that sentence structure, grammar and tenses are appropriate as there were a few sentences throughout the manuscript that came across slightly disjointed and did not flow as well.
Answer: We thank the reviewer comment´s:
We have carefully reviewed the manuscript and have restructured some sentences as appropriate so that there is connection and coherence in the content of the article.
